# Impacts of Male Intimate Partner Violence on Women: A Life Course Perspective

**DOI:** 10.3390/ijerph18168303

**Published:** 2021-08-05

**Authors:** Nerilee Hing, Catherine O’Mullan, Lydia Mainey, Elaine Nuske, Helen Breen, Annabel Taylor

**Affiliations:** 1School of Health, Medical and Applied Sciences, Central Queensland University, Bundaberg 4670, Australia; c.omullan@cqu.edu.au; 2School of Nursing, Midwifery and Social Sciences, Central Queensland University, Cairns 4870, Australia; l.mainey@cqu.edu.au; 3School of Arts and Social Science, Southern Cross University, Bilinga 4225, Australia; elaine.nuske@scu.edu.au; 4School of Business and Tourism, Southern Cross University, Lismore 2480, Australia; helen.breen@scu.edu.au; 5Queensland Centre for Domestic and Family Violence Research, Central Queensland University, Mackay 4740, Australia; a.taylor@cqu.edu.au

**Keywords:** violence against women, male partner violence, domestic violence, family violence, gendered violence, trajectories, transitions, turning points, thematic narrative analysis

## Abstract

The nature and extent of the impacts of intimate partner violence (IPV) on victims are well documented, particularly male partner violence against women. However, less is known about how these impacts might change over time, including their legacy after women leave an abusive relationship and the lasting effects in their later lives. The purpose of this study was to examine women’s experiences of IPV at different stages over their life courses. Interviews with a cohort of 18 older women who had left an abusive relationship were analysed using thematic narrative analysis and the findings were presented according to trajectories, transitions, and turning points over their life courses. When in the relationship, the women experienced direct impacts on their physical, mental, social, and financial wellbeing. During separation, many experienced continued abuse and housing, legal, and financial stress. Life after separation was marked by loneliness, trauma, financial insecurity, and damaged relationships. Some women reached a turning point in their recovery through helping others. Understanding these impacts can inform interventions during each stage. Crisis support is critical when women are in an abusive relationship and during the dangerous phase of separation. Interventions can also assist women’s longer-term wellbeing and help them recover through post-traumatic growth.

## 1. Introduction

Intimate partner violence is a major contributor to the burden of disease in women and a widespread public health issue [1,2]. Women are more likely than men to be physically, sexually, or emotionally abused by a current or former partner and to experience more severe impacts [3]. Numerous studies have examined the impacts of IPV on women victims. One systematic review concluded that “all aspects of victims’ lives were impacted” [4].

Impacts on mental health include elevated rates of depression, anxiety, post-traumatic stress disorder (PTSD), and suicide ideation [5,6,7]. Victimisation also increases the risk of physical health problems, including injury, poor functional health, chronic disorders, and traumatic brain injury [4,6,8,9]. IPV can contribute to victims’ health risk behaviours, including substance use [10,11] and gambling problems [12]. Their relationships are typically also affected, leading to low social connectedness, loneliness, and relationship dissatisfaction [4]. Victimisation can also harm women’s employment status and stability [13,14], and have negative financial consequences [15,16]. The nature and extent of the impacts of IPV on victims are well documented, particularly male partner violence against women.

However, less is known about how the impacts of IPV might change over time, including when women leave an abusive relationship and as their past experiences of abuse become more distant in time. In particular, the legacy of IPV on women’s lives after an abusive relationship ends, and that may endure into later life, has received relatively little research attention. Additionally, these longer-term impacts may change if women’s circumstances and wellbeing deteriorate or improve with age, or if certain factors enable or impede their recovery. Approximately half of women aged 45 years or over report being subjected to IPV at some point in their life [17]. Accordingly, understanding how the impacts of IPV change for women over time and in later life is important to inform appropriate interventions and support for a potentially sizeable cohort of women.

A life course perspective examines change over time [18] and therefore provides some useful tools for understanding changes in the impacts of IPV during women’s lives. A life course approach adopts a temporal and social perspective to look back across people’s experiences to help understand their current mental, physical, and social health [19]. This approach recognises that continuity and change influence people’s health across the life span and at different life stages, that contextual factors impact these experiences, and that earlier experiences shape later experiences, but that individuals can still exercise agency over their decisions and actions [20]. Several concepts are central to the life course perspective and were particularly informative in the current research. A *cohort* is a group of people born in the same historical period who share a distinctive social history that affects their attitudes, values, and behaviour throughout their life [21]. A *trajectory* is a long-term phase or pattern in a person’s life, such as a job or a marriage [22,23]. These relatively stable phases may be interrupted by *transitions*, which are short-term changes that occur when an old phase of life ends and a new one begins that involves changes in roles and status, such as the birth of children, separating from a partner, or remarrying [20]. *Turning points* may occur, constituting a marked and lasting shift in direction, which are brought about by life events that open or close new opportunities, bring about a lasting contextual change, or alter an individual’s self-concept, beliefs, or expectations [22,24].

Studies of IPV over one’s life course have focused mainly on patterns of violence, revealing how early experiences of violence are associated with negative health outcomes and the perpetration and victimisation of violence during adolescence and adulthood, as well as across generations [25,26]. Life course analyses also examined the temporal sequence of IPV and other aspects of health and wellbeing. For example, based on life course calendars, Yoshihama et al. [27] found that IPV victimisation negatively affected women’s contemporaneous employment and that cumulative IPV was associated with current and past-year health problems. In another example, Lindhorst and Oxford [28] found that adolescent exposure to IPV may alter the life course of young women, increasing their risk of IPV and mental health problems in adulthood. Other research has pointed to the long-term negative consequences of IPV on women’s mental and physical health, career and educational outcomes, and risk of subsequent abusive relationships [14,29,30].

Research also examined changes in IPV over the course of relationships and its impacts on women. A narrative study of older women who were abused throughout a long-standing intimate relationship identified different clusters of experiences over their lifespans based on changes in severity, escalation, visibility, and types of violence [31]. In contrast, other life course research has examined stability and change in violence across sequential intimate relationships. A large quantitative study [32] found that women who experienced severe multifaceted violence were most likely to transition out of relationships altogether, were somewhat likely to transition into new aggressive relationships, and were less likely to transition into non-violent relationships. Several factors can motivate women to transition out of abusive relationships, including a shift in perspective about themselves, their relationship, or their partner, as well as a heightened awareness about the effects on children, the dynamics of the abuse, its increasing severity, and options for support [33,34].

Studies have also examined changes over time after leaving an abusive relationship, the struggles in establishing a separate life, and moving towards healing and growth [35,36,37,38] For example, a large qualitative study [39] identified two main processes in IPV recovery: intrapersonal and interpersonal processes. Intrapersonal processes included participants regaining their identity, embracing new freedom over their life, healing, fostering acceptance, increased knowledge about IPV, considering the possibility of new intimate relationships, and acknowledging the long-term process of recovery. Interpersonal processes included building positive social support and using their experiences to help others. While not all women recover from IPV in this way [38], some women reach a turning point that results in post-traumatic growth [40].

As the above studies illustrate, life course theory provides a useful lens through which to examine changes over time in how women experience the impacts of IPV. The aim of the current study was to examine transitions, trajectories, and turning points in the experiences of IPV victimisation amongst a cohort of (now) older women, including any lasting impacts of past IPV in their later lives. Because this was an exploratory study, we did not seek to test specific hypotheses. However, based on the literature, we expected to observe changing patterns and impacts of IPV over the life course of the participants that can help to inform interventions that are appropriate to each life stage.

## 2. Materials and Methods

This analysis drew on interviews that were conducted for a larger study examining IPV against women, where the woman and/or her male partner had a gambling problem [41]. The interaction between gambling and IPV was detailed in that study and is of secondary interest in the current analysis. The current analysis was not conducted for the larger study.

### 2.1. Participants

Amongst the 72 women interviewed for the larger study, 18 women were aged 50 years or older and had separated from an abusive male partner. Focusing on this older cohort enabled us to analyse the reported impacts of IPV during the trajectories, transitions, and turning points that occurred during and after the abusive relationship, including into their later life. We did not ask the participants’ specific age but instead asked which age group they were in. A cutoff of 50 years was considered appropriate because 50–59 years and 60–69 years were the two oldest age groups amongst the participants. Twelve of the eighteen women had been subjected to IPV that was linked to their male partner’s gambling (WMG), with five of these participants focusing their interview mainly on his financial abuse (WFA). Six women had experienced IPV linked to their own gambling (WWG), where they patronised gambling venues to escape the abuse. The women were recruited through services that provided gambling help, domestic and family violence (DFV) support, and financial counselling, as well as advertisements that were placed online and in various forums. Ethics approval for the study was gained from the lead author’s institutional Human Research Ethics Committee. Before providing informed consent, the participants received an information sheet detailing the study’s purpose, what was involved for participants, assurances of anonymity, and contact details for help services. Table 1 shows the key characteristics of the participants. Twelve women were aged 50–59 years and six were aged 60–69 years. Thirteen women lived in a metropolitan location, while five lived in a regional area. Most participants lived in one of the three most populous states in Australia: New South Wales, Victoria, and Queensland.

### 2.2. Procedure

Four experienced researchers conducted unstructured telephone interviews lasting 30–120 min. Each participant was asked to tell her story of how IPV and gambling had impacted her life. Specifically, we asked, “I’m particularly interested in the role that gambling played in the IPV you have experienced. You might start from when problems first started occurring and tell me how things developed over time.” Using an unstructured approach allowed the women to focus on experiences and impacts of most importance to them. The interviewers had prompts to use (e.g., social support available, help-seeking), but found that these were rarely needed. All women said they were participating to help others and were therefore willing to share detailed narratives. We compensated each participant with an AU$40 gift card. All interviews were recorded with permission and professionally transcribed.

### 2.3. Analysis

To support the life course perspective, we used thematic narrative analysis to understand the participants’ constructed stories and how they interpreted their lived experiences [42,43]. This approach embeds prominent themes that are extracted from the narratives within the broader sequence of events. Because participants’ stories were necessarily subjective and interpretive, they provided a window into how they experienced events and the subjective meaning they ascribed to them [44]. We first composed a temporal sequence that reflected stages and patterns across participants’ life courses by mapping the trajectories, transitions, and turning points in each narrative. Second, thematic analysis drew out shared and contrasting elements across stories pertaining to the impacts of IPV on participants at these various stages [44,45,46]. Using an inductive approach, open coding of each interview transcript proceeded line by line, using an inclusive, thorough, and systematic process to identify the initial features that were of potential relevance to the experiences of the women participants. This coding was an iterative process, using the constant comparative method to add new codes, modify existing codes, and recode data as appropriate while working through the interview data. This process captured the diversity and patterns within the data. The coded data were then reviewed to identify areas of similarity and overlap between codes. Themes (categories) were generated by clustering or collapsing codes that shared some unifying features so that they captured a meaningful pattern in the data that was relevant to the research aim. Again, this was an iterative process of review and revision. Selective coding based on another review of the interview data was used to help saturate the themes with further evidence and data extracts (participant quotes). The credibility was enhanced by gathering data directly from participants with lived experiences, enabling participants to check their interview transcript, and adherence to the standard methods used in thematic analyses [47].

## 3. Results

This section describes the cohort of women and then gives the analysis results of the key elements in four sequential stages across the participants’ life courses:Trajectory 1: the abusive relationship and its shorter-term impacts;Transition: separation and its impacts;Trajectory 2: life after the abusive relationship;Turning point: healing through helping others.

### 3.1. The Cohort

The 18 women were aged 50–69 years. Many reflected on an upbringing by parents who conveyed traditional views of marriage, where the husband had elevated status as the breadwinner and decision maker, while the wife performed household duties. The women typically felt they were expected to marry and stay married, make sacrifices to solve any marital problems, obey their partner’s authority, and be “submissive” (WFA003). Due to being socialised into keeping marital problems private and with advice from their mothers that “if anything happens, keep it to yourself and try to fix it” (WMG007), some women recalled that they were “saving somebody else at my own expense” (WFA012) and thinking that “things will be okay if I make him happy” (WWG017).

The socialisation of many women to be dutiful left them vulnerable when they entered a relationship with an abusive partner. Consistent with the characteristics of abusive men across all age groups [48], the women described their partner as domineering, demanding, entitled, “manipulative” (WWG005), “a misogynist” (WMG078), and “restrictive and controlling” (WWG067). Men’s endorsement of traditional masculine ideology and gendered role expectations are consistently associated with male partner violence against women [49,50].

### 3.2. Trajectory 1: The Abusive Relationship and Its Shorter-Term Impacts

When in the relationship, all women experienced emotional and verbal abuse from their partner, ranging from denigration, blame, and gaslighting to screaming abuse, stalking, and death threats. This abuse often conveyed the man’s sustained disdain of his female partner, but also frequently arose when he lost at gambling or if she challenged him about his gambling:

“He got in the car and tried to run me over. He was driving on … the front lawn of the houses trying to run me over, like, for whatever reason it was my fault that … he’d done all his money and that he didn’t win at … gambling. So it was, like, well, someone has to pay.” (WMG002)

Approximately half the women were physically assaulted, including the partner punching, kicking, choking, or attempting to drown her, and trying to throw her out of a moving car. Some were sexually abused, including being drugged, raped, and videotaped. When one woman confronted her husband about stealing her money for gambling, he locked her and the children in the house, removed the phones, and terrorised them before killing their pets. Another woman related: “It was psychotic … he basically just punched me straight in the face as soon as he walked in the door. He accused me of having an affair” (WWG017).

All men with a gambling problem were reported to perpetrate financial abuse, including theft, fraud, creating debts in the woman’s name, selling the couple’s assets, redrawing on the mortgage without her knowledge, and spending all the family’s income on gambling. Financial abuse was typically accompanied by other forms of abuse: “Mine was a total combination of domestic violence, financial abuse, emotional abuse … he was just totally addicted to having a bet” (WFA027).

The shorter-term impacts of the women’s victimisation included physical injuries, weight loss, and exhaustion, and one woman described contracting sexually transmitted infections from her sexually abusive and philandering husband. One woman’s declining physical health resulted in her doctor advising her to: “leave the country and get away from him, because it’s killing you … honestly, in my view as your physician, I want you and the children to get away” (WWG061). Some of the women’s injuries required medical attention, including surgery for broken bones.

Impacts on the women’s mental health were starkly apparent and included loss of self-esteem from being constantly degraded and lied to, with most being “desperately unhappy” (WMG001). Reflecting their socialisation to be self-sacrificing and submissive in order to make the relationship work, many women blamed themselves for the problems and downplayed their own needs to try to meet their partner’s demands. Some women assumed personal responsibility for their partner’s abuse: “I must be doing something wrong. That’s why this is happening” (WMG007). They often felt that solving these problems required even more sacrifice:

“I kept thinking, which many women do: if you love more, if you give more, if you sacrifice more, then he will slowly come around that you are worthy and you’re a good enough woman.” (WMG078)

Stress, burnout, and mental exhaustion commonly resulted from living in fear, having to assume all household duties, and sometimes taking on extra employment to compensate for the gambling. Conversely, these pressures sometimes prevented women from maintaining their employment:

“You can’t do high-powered work and have two kids and run a family and deal with abuse. I just got … burnt out and I had to stop. And once the money stopped coming in, well, the problems just really escalated, really escalated, because he didn’t have the money to gamble with anymore.” (WFA006)

Some women reported serious trauma and became constantly fearful and withdrawn. One woman described her husband as being “threatening on a daily basis”, and therefore, she was constantly “in survival mode” (WMG001):

“I’d learnt not to do anything to make him angry, because it was frightening … he would go into rages … like a ferocious animal and I’d already become affected by the complex trauma, and so I just froze and became withdrawn. So, this was happening … cyclically … I was with him for a long time.” (WMG001)

The impacts of financial abuse were immediate and severe, including the inability to buy food, nappies, and other essentials. Some women were evicted because the family could not afford rent, and many were saddled with debt as they struggled to pay household expenses.

Nearly all women reported becoming socially isolated. Male partners often restricted the woman’s access to family and social engagements. Other women were “so immersed in … that world of danger … you don’t look up and out when you’re in that environment, you become quite insular” (WMG001). Women often avoided their family and friends due to shame, fear of negative judgment, or an expectation they would not be believed. Reflecting family and social expectations that marital problems remain private, the women faced substantial barriers to disclosing their abuse, which most often “stayed behind closed doors” (WFA027). Some reported that their family and friends stayed away as they did not want to get involved:

“I was too embarrassed to tell anybody really. I became quite withdrawn from the world. I lost quite a few of my girlfriends because it’s embarrassing. Them and their partners are going on holidays to the Gold Coast, and I’m sitting in a TAB [betting outlet] being abused.” (WFA027)

Some women responded to their abuse by gambling or drinking. This woman explained her unhappiness and the escape that her gambling and drinking provided:

“We lived for 10 years in separate rooms … I was so unhappy, and I was just doing my own thing. I was using the casino for an escape, to get away. So, I started into poker machines … I started to drink more … I was terribly, terribly unhappy.” (WWG005).

Clearly, the women in this cohort experienced numerous negative impacts from the abuse during this phase, including on their mental and physical health, finances, and social and family relationships. Nonetheless, the women identified numerous reasons for staying in the relationship, including traditional values of wanting the marriage to work, expectations of a fairytale marriage, and maintaining a two-parent family for the sake of the children: “I wanted a perfect marriage … My dream was that little family with the nice little house” (WWG024). Structural barriers, including a lack of safe housing, childcare, and financial support, along with a societal view of IPV as a private matter to be kept behind closed doors, were additional cohort effects that deterred these women from leaving the relationship.

### 3.3. Transition: Separation and Its Impacts

Two of the eighteen relationships ended when the man left for another woman, but female partners ended the remainder. A range of issues triggered this transition. Intensifying abuse, its cumulative effects, and reaching a breaking point formed a common pattern leading to the decision to separate. After years of abuse, this woman left after her partner took a week off work to gamble intensively, despite their financial stress; she recalled her anger and exhaustion after he failed to show up to family events, as he had promised:

“I went and found him at the TAB … I just got really brave and angry at the same time, and I thought, ‘No, I can’t do this anymore’ … in the safety of the TAB, I declared it was over.” (WFA006)

Other women also recalled “final straw” events. The following woman described the moment she decided to leave after her partner gambled the money for the children’s Christmas presents, absconded from a family holiday, and emptied the bank account so he could gamble, leaving no money for food or rent:

“There was a big shift in me after everything went out of the bank account … that was the last straw … I just freaked out, because I was already beyond stressed and I sort of broke down and screamed and cried … I made a decision then, the switch was flicked, and I’d shut off from him, that was it, I was done, yeah.” (WMG001)

Other women described how professionals helped them realise the danger posed by the escalating violence and that they urgently needed to leave. In one example, the male partner’s psychiatrist asked to see the woman separately and advised her “I don’t want you or your children with your husband, he has psychosis … I’m actually worried for your safety, and the kids” (WWG061). Another psychiatrist then instilled confidence in her to leave: “just remember, you’re strong and you’re sound, don’t listen to anything else” (WWG061). Services also assisted women to make practical arrangements to escape:

“[State legal service] who focused on domestic violence … Absolute life saviour … a half day intervention … outlining that I would be dead if I didn’t follow their instructions and do exactly what they needed me to do.” (WMG084)

For other women, family support proved instrumental in triggering and enabling their escape: “So, when my sister said to me, ‘Get out, come and live with me … I will support you for six months and we’ll work it out’, I got in the car and left” (WMG078). A few women described how their adult children assisted them to leave.

For many women, escaping their abuser took several years, extending the impacts of this transition stage. Some male partners refused to leave the house which belonged to the woman, wanting to stay to facilitate their gambling: “He was forced to leave by an intervention order, which he challenged over a period of five years … he was on a gravy train … and didn’t want to get off” (WFA006). Some male partners terrorised and stalked the woman after she ended the relationship, with some women compelled to move into a refuge because he would find her if she stayed with family or friends:

“I couldn’t get rid of him, so I had to leave that place and move … living with my family in someone else’s house … that’s when he was getting really, really, really, really angry … when he started stalking ... I suffered for nearly six years ‘til … he went to jail … for stalking, for deprivation of liberty too. He was under my house with a knife, and I was his captive for three and half to four hours.” (WMG007).

Other women were abused and threatened by their partner during this period of transition: “From the day I declared it over up until the intervention order was served on him, he threatened to kill me every single day” (WFA006).

Women also described financial impacts during the transition period while they waited for court settlements. Financial impacts were particularly acute where the male partner had a gambling problem and continued to drain the woman’s finances:

“He dragged me through the family law system over a period of five years and … just continued to gamble and drive himself further into debt … whatever he gambled was my loss as well.” (WFA003)

This transition stage was also prolonged for some women because services were not always supportive. Reflecting the more limited understanding of IPV at the time, some health professionals blamed the women for the abuse, with one doctor saying “Well, you probably deserved it” (WWG003). Some women recalled that the police treated them as annoying hysterical women, failed to protect them, discouraged them from laying charges, and did not take the abuse seriously.

In summary, women described a range of triggers for leaving their abusive partner, including anger, exhaustion, and decisiveness after a “last straw” event, and receiving support from services and family, although not all services were supportive. However, the IPV had ongoing impacts during this transition because of their ex-partner’s continued violence, housing stress, financial stress, added debt, and lengthy legal processes.

### 3.4. Trajectory 2: Life after the Abusive Relationship

The women described how the IPV had cumulative and prolonged effects that lasted into later life on their mental health, finances, relationships, and gambling.

#### 3.4.1. The Legacy of IPV for the Women’s Mental Health

The longer-term impacts of IPV were very apparent on the women’s mental health. As one woman bluntly put it: “That’s why I ended up with complex trauma” (WMG001). Another woman who had left an abusive relationship ten years earlier commented: “To this day, it still has scars. It’s an ongoing battle” (WWG017). Another participant spoke of lifelong effects, that her partner’s abuse and gambling “ruined my whole life … you spend the rest of your life making up for what was happening at that time” (WFA027). PTSD could result from sustained abuse and continue to be debilitating:

“There are moments where, you know, you’re going to be overcome with grief and pain. I mean there are days when I cry, because grief is a very volatile emotion, and there are days where the grief washes over me. There are days when I just can’t do any work, or I just can’t function properly, because that’s part of the post-traumatic stress disorder … You’ve just got to … recognise the pain and the grief, you can’t deny it.” (WFA006)

Years of abuse had eroded other women’s self-esteem and identity and they continued to battle. One woman explained that the distress remained “overwhelming. I guess the strategy I take is one day at a time and make the most of each day” (WFA012). The legacy of the abuse on many women’s mental health was debilitating long after the relationship ended.

#### 3.4.2. The Legacy of IPV for the Women’s Finances

All women whose partners had gambled heavily experienced long-lasting financial damage. Women who had partnered later in life often brought substantial assets to the relationship and had an established career and a good income. Therefore, they had much to lose from their partner’s gambling and financial abuse. One woman’s partner re-mortgaged the house she had nearly paid off so he could pay his gambling debts:

“He has eroded … the assets I actually had acquired before I met him … I came into this marriage with 80 per cent more…of that particular house than I’ve gotten out of it … I deeply resent my assets going towards paying off his debts.” (WFA003)

Women spoke of being drained of all the assets that they accumulated over a lifetime of work and their limited capacity to recover during the remainder of their working life: “I lost pretty much everything that I’d worked my entire life towards” (WFA006). One woman could no longer gain employment in her sector because she had a police record as her ex-partner had taken out a DVO against her in retaliation:

“I’m worse off … I’ve lost my money, and I’ve lost my ability to get a job … I’m still in debt … I’ve got no assets … and I’ve got like 10 years before I retire.” (WFA012)

It could take a long time to separate financial affairs from their ex-partner, and women could face the sole financial burden of childcare because the partner refused to shoulder any responsibility. These men continued to gamble and financially abuse their former partners after separation:

“I’m still bankrolling him … We’ve been forced to co-parent because … we have equal shared responsibility, and there is nothing equal, shared, or responsible about an abusive gambler. So, I continue to carry him, and I continue to carry his debts. He hasn’t honoured any of the court orders … things that he’s been told to pay and cover, he didn’t pay, and so that puts the burden on the responsible party, doesn’t it?” (WFA006)

The financial situation of women is already unequal across the life course as they tend to have lower-paying jobs and less superannuation. The financial impacts of IPV further undermine their long-term needs.

#### 3.4.3. The Legacy of IPV for the Women’s Relationships

The women’s earlier experiences of IPV had lasting impacts on their relationships. Some were reluctant to enter new romantic relationships because their trust and self-esteem had been so diminished by the abuse:

“Still to this day I’m single … I’ve never really trusted anyone. I push people away … Your self-worth becomes quite low … You think you’re not good enough for them.” (WFA027)

Many women spoke of damaged relationships with children, family, and social networks that lasted for long periods or their entire lives. One woman’s sister refused to talk to her for 10 years until she left her abusive partner. Relationships with children could be irreparably damaged if the children blamed their mother for exposing them to prolonged violence:

“My daughter … screamed her head off at me and said ‘I hate you. I hate you’ … shot me full of holes because she said I should have left so that they never had to see that.” (WMG002)

Some ex-partners actively worked to turn the children against the woman; modelled disrespectful attitudes towards her, which the children copied; or undermined their relationships with grandchildren:

“He used to manipulate my daughter and now he’s manipulating my grandson and that is family violence and it’s very, very psychologically damaging for my grandson to think that I would be harming him.” (WMG078)

Even though these women may have been abused many years ago, their earlier experiences continued to impact their relationships with friends and family into later life.

#### 3.4.4. The Legacy of IPV for the Women’s Gambling

Many women remained unpartnered and several gambled to escape loneliness and boredom as they aged. The “empty nest syndrome” contributed to ennui and depression, with these women feeling they were no longer valued or had a useful role. Gambling venues were some of the few places where they could feel comfortable, safe, and have some social contact, because “as you get older, it’s harder to make new friends. Like, where do you go?” (WWG024). Some women gambled to delay coming home to an empty house:

“God I’ve got to go home to an empty house, I just don’t think I can take that, I need to go somewhere where I can just sit and have a drink, relax … you think, god, you’ve had this whole life … and here you are on your own, what’s wrong with you? … the loneliness-that’s a killer, just after all of that, you’re on your own … You could say I’m just empty, yeah … I still feel incredibly fragile and broken … there’s a big void, and it’s lonely.” (WWG061)

The impacts of IPV on some women’s mental health also drew them to playing gaming machines as they provided “a dead zone … to step out of everything” (WWG061), an emotional escape from past trauma, present worries, and low self-worth:

“My self-esteem was so low I actually felt the need to punish myself … for allowing myself to be in that position … I would punish myself financially until I basically ran out of money.” (WWG017)

Remaining single in their older age because of their past victimisation led some women to gamble in later life to help cope with loneliness, boredom, social isolation, and their diminished mental health.

### 3.5. Turning Point: Healing through Helping Others

Some women in this cohort had reached a turning point in their lives, shifting focus to helping others affected by IPV or gambling. Reaching this point involved being able to name their past abuse as IPV, reflect on their lived experience, and be willing to share their story. All women noted that their desire to help others motivated their participation in the study: “And now I just want to give back … I can truly help from my experience” (WMG005).

Several women now worked in family violence support agencies. One participant explained that her lived experience was invaluable in her work with women and children in gambling and DFV situations, and in running a men’s behaviour change program:

“I do bring a wealth of experience for the victim/survivors and when I deal with men … I do it for the women and children behind them … I bring that context of gambling and family violence into my behaviour change program.” (WMG078)

Some women worked with charities, using social media, community speaking, and advocacy to break the silence around DFV and gambling. One woman who had established a not-for-profit association to advocate for gambling reform provided crisis support. Another woman explained her advocacy and community education work:

“I speak on the topic of family violence … I’m a trained advocate with [a family violence response service], and I’ve done quite a lot of presentation work with the Financial Services Ombudsman … I’ve participated in the making of a video for the DV sector on strength and resilience … presented to numerous politicians on DV.” (WFA003)

Some women contributed to community education and professional development by sharing their personal stories at schools, conferences, theatrical performances, and services. The women who had reached this turning point recognised that sharing their story, actively helping others, and advocating for change assisted with their own recovery, personal growth, and increased empowerment: “it helps your own healing” (WMG001). One woman had written a book to help rebuild her confidence:

“If I keep it to myself, it means I believe something is wrong, and I want to have my confidence back. That’s why I opened up and I said, ‘Now, I have to tell my story, to help other people’.” (WMG007)

Many women in this cohort spoke about eventually drawing on their lived experience to help others facing IPV and/or gambling issues. This turning point was a marked shift in their life course trajectory that involved a move towards healing and self-empowerment.

## 4. Discussion

This study used a life course perspective to explore the impacts of IPV on a cohort of (now) older women during an abusive relationship, in transitioning out of the relationship, after separation, and for women who had reached a turning point. Impacts on the women’s physical, mental, social, and financial wellbeing differed during each stage and could inform the most useful types of support in each phase.

While in the abusive relationships, the most immediate threats were to the women’s physical safety. This was exacerbated by a lack of effective support at that time, with the women reporting victim-blaming, being stereotyped as over-emotional and irrational, and a failure to take their abuse seriously. Effective police responses, safety plans, and active bystander responses are critical at this time to help protect women from further violence [51]. Services providing crisis support also need to prioritise the woman’s safety [52]. Given the variation in women’s situations and responses to IPV, therapeutic support should be attuned to the woman’s readiness to change [53,54,55] to help her move from the pre-contemplative to contemplative stage of change [56]. Appropriate support can help women to recognise the abuse and its risks and evaluate the pros and cons of changing their situation to help them move towards a decision that is in their best interests [54,57,58].

However, few women seek help while in an abusive relationship due to shame, self-blame, concerns about victim-blaming, and fear of further violence [59]. As was found in this study, living in an abusive relationship depletes women’s mental health; they are typically anxious, emotionally exhausted, deeply unhappy, and may lack the emotional strength to disclose their situation [4,6,7]. These women are therefore in critical need of social and family support to reduce their vulnerability, help them cope, and to provide practical help. However, women in this study were usually socially isolated and estranged from extended family due to shame about the situation, their diminished self-esteem, and their partner’s control over their independence. Having been socialised into traditional family values, where women were expected to assume responsibility for maintaining a happy family, meet their partner’s demands, and keep marital problems hidden, these women often blamed themselves and suffered in silence. Community education can encourage family and friends to offer non-judgmental support to victims rather than avoid or shame them, and to support them to seek professional help. This may include help from family violence, mental health, gambling help, financial counselling, legal, and other services. The women with a problem gambling partner all experienced financial abuse. Financial institutions need to be able to assist these women to prevent ongoing financial damage.

Often after years of maintaining the façade of a perfect marriage, as expected at the time, the women’s decisions to leave the abusive relationship were triggered by numerous factors. These included escalating violence, a “final straw” event, or help from professionals, family, or friends. As was found in other research, escalating violence may prompt the woman to realise that her partner’s behaviour is unlikely to change and poses a real risk of lethality, bringing her closer to a decision to leave [52,57,60]. Professional help can then assist her to progress to the preparation and action stages of change [56], including accessing necessary resources and protections to optimise her and her children’s safety [57]. Some women in this study were assisted by services to implement a safe escape plan, while others were supported by practical and emotional support from family and friends. However, reflective of the situation several years ago, many women found that services, family, and friends had little understanding of IPV and that there were few practical sources of support available, which tended to prolong the relationship and the women’s experiences of violence.

Transitioning out of an abusive relationship is a particularly dangerous time [51,61], with some women in the current research reporting ongoing threats, violence, and stalking once they left. Safety issues are paramount and require prompt and protective police actions to keep the abuser away, including domestic violence orders [52]. Crisis support is also critical, including safe refuges, emergency funds, and help from friends and family [57,58]. Professional help is also needed, not only to assist with the psychological challenges that many women experience but also the practical and legal issues involved in separation. Many women indicated that legal processes were extremely slow and allowed for the perpetrator’s continued manipulation and abuse. Improved processes would allow women to limit this interaction and move more quickly towards a new life phase.

The women’s narratives revealed impacts on their mental health, relationships, and finances that endured long after the abusive relationship. Current responses to IPV focus on crisis support, with less attention given to longer-term support. Many IPV victims experience PTSD, often accompanied by depression, suicidal behaviours, and substance abuse [62]. Women with long-term trauma have complex needs that require services that can provide multidisciplinary integrated responses, such as developmental-relational, cognitive-behavioural, skill-building, and strengths-based approaches [63]. Strengths-based approaches and intentional activities may particularly assist older women with depressive symptoms by cultivating positive feelings, behaviours, and cognitions [64]. Strategies are also needed to combat the social isolation and boredom that some women experience after an abusive relationship, such as providing safe social spaces as alternatives from gambling venues.

Several women had reached a turning point in their recovery, representing a distinct shift in their life course trajectory. These women appeared to have an increased sense of purpose, renewed personal strength, meaningful roles, and changed priorities, which reflected post-traumatic growth [40]. Numerous women spoke about experiencing meaningful personal change and the therapeutic value of drawing on their lived experiences to help others. Having a renewed purpose, social activism, and public narration of personal tragedy and transition are heavily implicated in post-traumatic healing [65,66]. These findings suggest that supporting IPV survivors to help others, raise public awareness of IPV, and advocate for change are valuable interventions to assist women’s post-traumatic growth and help to maintain their continued recovery [67].

Like all research, this study has limitations. The sampling prioritised the willingness to share rich accounts of lived experience over generalisability. Most participants had been victimised many years earlier, which may have hindered recall. However, the passage of time enabled them to recognise how their experiences had impacted their life. All women had received professional support for the IPV and/or gambling. This may have helped their reflexivity such that women who have not sought help may experience or report the impacts of IPV differently. While participants came from a variety of socioeconomic, cultural, and educational backgrounds, the small sample size did not allow for analyses of these intersectional issues. All women’s experiences of IPV were related to their or their partner’s gambling; the impacts of IPV that are not linked to gambling may differ. The interviews focused on the relationship between gambling and IPV, so some impacts of IPV may not have been disclosed. Further, the study was unable to apply a full life course health development approach that accounts for all influences on health outcomes [68].

Given that this study may be considered an exploratory pilot study due to its limited sample size, research with larger samples is needed to confirm the findings and trace generalisable profiles of trajectories, transitions, and turning points for women who have been victimised by IPV. Research using life course calendars, as well as prospective studies, could be valuable in this regard. Of interest is whether the life course experiences found in this study remain consistent in larger samples, as well as in samples of women who have not sought professional help and those of varying cultural, socioeconomic, health, and educational backgrounds.

## 5. Conclusions

This study contributes to the relatively scant knowledge on the changing patterns and longer-term impacts of IPV over the life course, particularly by presenting rich qualitative insights to elaborate themes that were highlighted in previous research. Its main contribution was to emphasise that the impacts of an abusive intimate relationship can last a lifetime and vary in nature and severity during different stages of the life course. When in the relationship, women experienced direct impacts on their physical, mental, social, and financial wellbeing. During separation, many experienced continued abuse and housing, legal, and financial stress. Life after separation was marked by loneliness, trauma, financial insecurity, and damaged relationships. Some women reached a turning point in their recovery through helping others. Understanding these impacts can help to inform interventions during these different trajectories, transitions, and turning points to assist women victims of IPV to deal with these immediate and lasting impacts.

## Figures and Tables

**Table 1 ijerph-18-08303-t001:** Key characteristics of the participants.

No.	Participant ID	Age Group (Years)	State	Location
1	WMG001	50–59	New South Wales	Metropolitan
2	WMG002	60–69	Queensland	Regional
3	WMG005	50–59	Tasmania	Metropolitan
4	WMG007	60–69	Victoria	Metropolitan
5	WMG008	50–59	Queensland	Regional
6	WMG078	50–59	Victoria	Metropolitan
7	WMG084	50–59	Queensland	Metropolitan
8	WWG003	60–69	Victoria	Regional
9	WWG017	50–59	Queensland	Metropolitan
10	WWG024	50–59	Victoria	Metropolitan
11	WWG045	50–59	South Australia	Metropolitan
12	WWG061	60–69	Queensland	Metropolitan
13	WWG067	50–59	New South Wales	Regional
14	WFA003	60–69	Queensland	Metropolitan
15	WFA006	50–59	Victoria	Metropolitan
16	WFA012	50–59	New South Wales	Metropolitan
17	WFA021	60–69	Queensland	Regional
18	WFA027	50–59	New South Wales	Metropolitan

## Data Availability

Data are not available on request or publicly due to privacy and ethical restrictions, in order to protect the participants’ safety.

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
