# Peer review of "Impacts of Male Intimate Partner Violence on Women: A Life Course Perspective"

_ijerph, 2021, doi:10.3390/ijerph18168303_

Round 1
Reviewer 1 Report
This is a valuable piece of research focussing on IPV in relation to gambling as its second interest. The methodology is sounds, results are properly presented. However, the authors might work on having a better link between their cohort and trajectories' values: while the cohort under study was described in 3.1.1, there was no sufficient follow-up through trajectories, transition and turning points as well as no reflection in the discussion and conclusions.
Author Response
Thank you for this valuable advice. We have now linked the values of the cohort more explicitly to their experiences, especially the values that tended to keep the women in the relationship for longer and to keep the IPV private, which extended their experiences of violence. We have also incorporated these issues into the Discussion section.
Reviewer 2 Report
This study captured the perspectives of 18 older women who had left an abusive relationship. The study appeared to be well done. However, the main concern with this study is that it did not seem to break new ground. Certainly, replication studies are important and this study may be an important one in Queensland, but it did not appear to provide new information or different perspectives from that which is already known. Therefore, even though this is a qualitative study that could potentially have provided new perspectives and new insights, its primary importance seems to be that the themes were somewhat more elaborated because of the qualitative nature of the study.
Still, the study was well done and the paper was well written and interesting.
Author Response
Thank you for your feedback. We have now acknowledged that a key contribution of the study has been to provide rich qualitative insights to elaborate themes that have also been highlighted in previous research.
Reviewer 3 Report
The manuscript details an interesting study focusing on an investigation of the impact of intimate partner violence on female partners using the life course theoretical lens. This qualitative study is well-written and structured. The point of departure from similar studies is well articulated enabling the reader to easily delineate between the study and studies focusing on similar themes. The choice of the methods for data collection and analysis are robustly justified. The use of thematic narrative analysis for analysing the interviews with the selected cohort is considered apt for the intended purpose.
The manuscript provides a succinct discussion section wherein comparisons of the study's findings with findings from similar studies are clearly highlighted. The implications of the study's findings as well as the apposite nature of the life course theory for unravelling this phenomenon are expressly provided in the manuscript. The conclusions of the study are congruent with its original aim.
Author Response
Thank you for your positive feedback. No changes have been requested by this reviewer.
Reviewer 4 Report
The present paper intends to longitudinally address the consequences of intimate partner violence for women who scaped from this situation at some point in their lives. This is a quite relevant topic these days as gender-based violence is an important problem in our societies. Despite the significance of the research, some aspects could be improved.
First, authors should include the aim of the study and hypotheses. However, sample is rather small (only 18 women). I suggest authors to increase their sample. Besides, authors should provide complete information about the characteristics of the sample.
As the whole research is based in the interviews, more information about the characteristics of the interview and the way that participants responses were systematically coded should be provided.
The qualitative nature of the study toghether with the relative small sample size impairs to trace generalizable profiles. Thus, this study should be viewed as an exploratory pilot study. This should be discussed by authors as well as future direction based on their own results.
Author Response
Thanks you very much for your feedback and helpful suggestions.
The aim of the study was articulated in lines 120-123. We have now changed “The current study aimed…” to “The aim of the current study was…” to help this stand out more clearly.
We have not included any hypotheses, as hypotheses are not usually presented for qualitative research with a small sample. As the reviewer rightly notes, this should be seen as an exploratory pilot study and we have now noted this in the Discussion, along with directions for future research based on the results.
Unfortunately, we are unable to increase the sample size as the study period is now complete, but we note that small information-rich samples are the usual practice in qualitative research, which does not seek to generalise the results to the population.
We have now included Table 1 which shows the key characteristics of the participants.
We have also provided more details about the characteristics of the interview, explaining that we specifically asked: “I’m particularly interested in the role that gambling played in the IPV you have experienced. You might start from when problems first started occurring and tell me how things developed over time.”
We have also provided more detail about how the responses were systematically coded.
To address the last point raised by the reviewer, we have now added: “Given that this study may be considered an exploratory pilot study, due to its limited sample size, research with larger samples is needed to confirm the findings and trace generalizable profiles of trajectories, transitions and turning points for women victimised by IPV. Research using life course calendars, as well as prospective studies, could be valuable in this regard. Of interest is whether the life course experiences found in this study remain consistent in larger samples, as well as in samples of women who have not sought professional help and those of varying cultural, socioeconomic, health and educational backgrounds.
Reviewer 5 Report
This study presents an interesting qualitative examination of the impacts of intimate partner violence (IPV) on victims over times. One major concern is that authors need to clarify why the age cutpoint 50 was adopted when selecting the eligible participants.
One minor point is that the Conclusion can be more precise, aiming to conclude/summarise the primary research question raised.
Author Response
We have now explained why age 50 was the cutoff point when selecting the eligible participants. We have also focused the Conclusion more explicitly on the summarised results which address the overall aim of the study, and trimmed its length where possible.
Round 2
Reviewer 4 Report
Authors have made a considerable effort in addressing the comments of the reviewers and the paper has improved enough for publication. I thank authors for the job. However, I would suggest to replace content from lines 122 - 125 from a temptative hypothesis or hypotheses regarding the literature review. These sentences are more appropiate for a conclusion.
Author Response
Thank you for these further comments. We have now addressed these by changing lines 120-125 to read "The aim of the current study was to examine transitions, trajectories and turning points in the experiences of IPV victimisation amongst a cohort of (now) older women, including any lasting impacts of past IPV into their later life. Because this was an exploratory study, we did not seek to test specific hypotheses. However, based on the literature we expected to observe changing patterns and impacts of IPV over the life course of the participants that can help to inform interventions that are appropriate to each stage."